# Red Blood Cells from Individuals with Lesch–Nyhan Syndrome: Multi-Omics Insights into a Novel S162N Mutation Causing Hypoxanthine-Guanine Phosphoribosyltransferase Deficiency

**DOI:** 10.3390/antiox12091699

**Published:** 2023-08-31

**Authors:** Julie A. Reisz, Monika Dzieciatkowska, Daniel Stephenson, Fabia Gamboni, D. Holmes Morton, Angelo D’Alessandro

**Affiliations:** 1Department of Biochemistry and Molecular Genetics, University of Colorado Anschutz Medical Campus, Aurora, CO 80045, USA; julie.haines@cuanschutz.edu (J.A.R.); monika.dzieciatkowska@cuanschutz.edu (M.D.); daniel.stephenson@cuanschutz.edu (D.S.); fabia.gamboni@cuanschutz.edu (F.G.); 2Central Pennsylvania Clinic, A Medical Home for Special Children and Adults, Belleville, PA 17004, USA; dholmesmorton@gmail.com

**Keywords:** Lesch–Nyhan, purine salvage, hypoxanthine-guanine phosphoribosyltransferase deficiency, metabolomics, redox proteomics, lipidomics

## Abstract

Lesch–Nyhan syndrome (LN) is an is an X-linked recessive inborn error of metabolism that arises from a deficiency of purine salvage enzyme hypoxanthine-guanine phosphoribosyltransferase (HPRT). The disease manifests severely, causing intellectual deficits and other neural abnormalities, hypercoagulability, uncontrolled self-injury, and gout. While allopurinol is used to alleviate gout, other symptoms are less understood, impeding treatment. Herein, we present a high-throughput multi-omics analysis of red blood cells (RBCs) from three pediatric siblings carrying a novel S162N HPRT1 mutation. RBCs from both parents—the mother, a heterozygous carrier, and the father, a clinically healthy control—were also analyzed. Global metabolite analysis of LN RBCs shows accumulation of glycolytic intermediates upstream of pyruvate kinase, unsaturated fatty acids, and long chain acylcarnitines. Similarly, highly unsaturated phosphatidylcholines are also elevated in LN RBCs, while free choline is decreased. Intracellular iron, zinc, selenium, and potassium are also decreased in LN RBCs. Global proteomics documented changes in RBC membrane proteins, hemoglobin, redox homeostasis proteins, and the enrichment of coagulation proteins. These changes were accompanied by elevation in protein glutamine deamidation and methylation in the LN children and carrier mother. Treatment with allopurinol incompletely reversed the observed phenotypes in the two older siblings currently on this treatment. This unique data set provides novel opportunities for investigations aimed at potential therapies for LN-associated sequelae.

## 1. Introduction

Lesch–Nyhan (LN) syndrome is a disease that affects approximately one in every three-hundred-and-eighty-thousand births. Over 2000 known genetic mutations contribute to the etiology of LN, a rare X-linked recessive disorder [1,2]. A wealth of mutations have been described within the intronic or exonic regions 1–9 on chromosome Xq26-27, genetic abnormalities that manifest biochemically with the deficient activity of hypoxanthine-guanine phosphoribosyltransferase (HPRT). HPRT is a purine salvage enzyme responsible for the conversion of hypoxanthine to IMP and guanine to GMP. As such, deficient HPRT activity results in the accumulation of hypoxanthine, feeding further oxidation by xanthine oxidase to urate, with concomitant excess generation of hydrogen peroxide, ultimately leading to the development of gout [3].

Clinical manifestations of the disease are severe and heterogenous; symptoms can include intellectual deficits, reduced gray and white matter in the brain, self-injurious behavior, hypercoagulability, macrocytic anemia, and gout [4,5]. With X-linked recessive genetic etiology, only males develop LN disease, though females may be heterozygous and, in some cases, display mild symptoms. While advances in the understanding of the mechanisms leading to gout have informed interventions with xanthine oxidase-targeting molecules, like allopurinol, further advancements in the understanding of the molecular manifestations of this disease are necessary to advance more successful interventions beyond the treatment of gout [6].

The breakdown and deamination of adenosine triphosphate to hypoxanthine is a hallmark of hypoxia-induced mitochondrial dysfunction [7,8], under either environmental or pathological conditions (e.g., ischemic or hemorrhagic hypoxia) [9]. In vivo, hypoxia is counteracted by red blood cells (RBCs), the dominant cell type in the blood and the most numerous cell type in the human body (~84% of total cells in an adult individual) [10]. Circulating RBCs are tasked with oxygen carrying and delivery, a process they have evolved to fulfill by losing organelles (including nuclei and mitochondria) to maximize hemoglobin content. These molecular adaptations are accompanied by the incapacity of RBCs to synthesize new proteins, for example, to replace oxidatively damaged components in the face of ongoing Fenton and Haber–Weiss chemistry as a result of the excess load of oxygen (up to 1 billion molecules per cell) and oxygen-coordinating iron in a mature erythrocyte [10]. In this sense, RBCs are extremely sensitive to perturbations in metabolic and redox homeostasis: For example, lacking mitochondria, RBC exposure to oxidant stress in vivo or ex vivo promotes purine deamination via AMP deaminase 3, which in turn favors the accumulation of inosine monophosphate, a precursor to hypoxanthine [11]. Ultimately, elevated RBC hypoxanthine is associated with altered morphology and increased extravascular hemolysis via splenic sequestration and erythrophagocytosis [12].

The goal of the Lesch–Nyhan study described herein is to provide metabolic insights that could inform effective treatments through dietary and/or metabolic interventions. A comprehensive molecular profiling of RBCs from LN patients has not previously been described and is the focus of the present study. Herein, we enrolled a family of five with the goal of leveraging a multiomics approach to describe in detail, for the first time, the molecular derangements in RBCs from three male LN patients as compared to the carrier mother and the clinically healthy father.

## 2. Materials and Methods

### 2.1. Subject Recruitment and Sample Collection

Blood samples were collected through venipuncture from a family of volunteers, including father (clinically healthy), mother (carrier), and three male children diagnosed with Lesch–Nyhan syndrome at the Central Pennsylvania Clinic and at Boston Children’s and Lancaster General Hospitals under institutionally reviewed protocols (No. 2014-12) and upon the signing of informed consent. The children were each homozygous for the HPRT1 c.485 G>A;p.Ser162Asn variant and presented with severe disability from generalized dystonia, self-injury, and increased serum uric acid. At the time of blood collection, their ages were 4 years, 2.5 years, and 4 months old. Owing to the very young age, only the two older siblings were being treated with allopurinol (mean dose 6.44 mg/Kg of weight per day) [13]. RBCs were separated from whole blood through centrifugation for 10 min at 4 °C and 2000× *g*.

### 2.2. High Throughput Metabolomics

Metabolomics analyses were performed as previously described [14]. Red blood cell samples were thawed on ice, then a 10 μL aliquot was treated with 90 μL of ice cold 5:3:2 MeOH:MeCN:water (*v*/*v*/*v*) then vortexed for 30 min at 4 °C. Supernatants were clarified by centrifugation (10 min, 12,000× *g*, 4 °C). The resulting metabolite extracts were analyzed (10 μL per injection) using ultra-high-pressure liquid chromatography coupled to mass spectrometry (UHPLC-MS—Vanquish and Q Exactive, ThermoFisher, Bremen, Germany). Metabolites were resolved on a Phenomenex Kinetex C18 column (2.1 × 150 mm, 1.7 μm) at 45 °C using a 5-min gradient method in positive and negative ion modes (separate runs) over the scan range 65–975 *m*/*z* exactly as previously described [14]. Oxylipins were resolved on a Waters ACQUITY UPLC BEH C18 column (2.1 × 100 mm, 1.7 μm) at 60 °C using mobile phase (A) of 20:80:0.02 MeCN:water:formic acid (FA) and a mobile phase (B) of 20:80:0.02 MeCN:isopropanol:FA. For negative mode analysis, the chromatographic gradient was as follows: 0.35 mL/min flowrate, 0% B 0–0.5 min, 25% B at 1 min, 40% B at 2.5 min, 55% B at 2.6 min, 70% B at 4.5 min, 100% B at 4.6–6 min, and 0% B at 6.1–7 min. The Q Exactive MS was operated in negative ion mode, scanning in full MS mode (2 μscans) from 150 to 1500 *m/z* at 70,000 resolution, with 4 kV spray voltage, 45 sheath gas, and 15 auxiliary gas. Following data acquisition, .raw files were converted to .mzXML using RawConverter version 1.2.0.1, then metabolites were assigned and peaks were integrated using Maven (Princeton University, Princeton, NJ, USA) in conjunction with the KEGG database and an in-house standard library. Quality control was assessed as using technical replicates run at the beginning, end, and middle of each sequence as previously described [14].

### 2.3. Transition Metal Analysis

Transition metals ^23^Na, ^24^Mg, ^39^K, ^44^Ca, ^57^Fe, ^63^Cu, ^77^Se, and ^66^Zn were measured using inductively coupled plasma–mass spectrometry on a Thermo iCAP RQ ICP-MS coupled to an ESI SC-4DX FAST autosampler system using 10 μL of RBCs exactly as previously described [15]. Final dilutions of 1:250 and 1:3750 were then analyzed via ICP-MS. Different dilutions were used to ensure that all analytes fell within the calibration curves (selenium reads fell just below calibrant 1; however, all intensities were approximately 10× above the blanks, so values were extrapolated). All chemicals and materials used for ICP-MS analysis were obtained from ThermoFisher, Waltham, MA, USA; ICP-MS calibrants and solutions were obtained from SPEX CertiPrep. Instrument performance was assessed by infusing an internal standard mix via a peristaltic pump and monitoring signal throughout the run. Additionally, separate quality controls of a known concentration (75 ppb) of each analyte were injected at the beginning, throughout the run between samples, and at the end of the run. The acceptance criterion for all QCs was ±25% of the known concentration. Thermo Scientific Qtegra software version 2.10.3324.131 was used for all data acquisition and analysis.

### 2.4. Untargeted Lipidomics

Total lipids were extracted as previously described [16]: 10 μL of RBCs were mixed with 90 μL of cold methanol. Samples were then briefly vortexed and incubated at −20 °C for 30 min. Following incubation, samples were centrifuged at 12,700 RPM for 10 min at 4 °C and 80 μL of supernatant was transferred to a new tube for analysis. Lipid extracts were analyzed (10 μL per injection) on a Thermo Vanquish UHPLC/Q Exactive MS system using a 5-min lipidomics gradient and a Kinetex C18 column (30 × 2.1 mm, 1.7 µm, Phenomenex) held at 50 °C. Mobile phase A: 25:75 MeCN:water with 5 mM ammonium acetate; Mobile phase B: 90:10 isopropanol:MeCN with 5 mM ammonium acetate. The gradient and flow rate were as follows: 0.3 mL/min of 10% B at 0 min, 0.3 mL/min of 95% B at 3 min, 0.3 mL/min of 95% B at 4.2 min, 0.45 mL/min 10% B at 4.3 min, 0.4 mL/min of 10% B at 4.9 min, and 0.3 mL/min of 10% B at 5 min. Samples were run in positive and negative ion modes (both ESI, separate runs) at 125 to 1500 *m*/*z* and 70,000 resolution, 4 kV spray voltage, 45 sheath gas, and 25 auxiliary gas. The MS was run in data-dependent acquisition mode (ddMS^2^) with top10 fragmentation. Raw MS data files were searched using LipidSearch v 5.0 (ThermoFisher).

### 2.5. Global Proteomics

Proteomics analyses were performed as described [17]. A volume of 10 μL of RBCs were lysed in 90 μL of distilled water. Then, 5 μL of lysed RBCs were mixed with 45 μL of 5% SDS and then vortexed. Samples were reduced with 10 mM DTT at 55 °C for 30 min, cooled to room temperature, and then alkylated with 25 mM iodoacetamide in the dark for 30 min. Next, a final concentration of 1.2% phosphoric acid and then six volumes of binding buffer (90% methanol; 100 mM triethylammonium bicarbonate, TEAB; pH 7.1) were added to each sample. After gentle mixing, the protein solution was loaded to an S-Trap 96-well plate, spun at 1500× *g* for 2 min, and the flow-through collected and reloaded onto the 96-well plate. This step was repeated three times and then the 96-well plate was washed with 200 μL of binding buffer three times. Finally, 1 μg of sequencing-grade trypsin (Promega, Madison, WI, USA) and 125 μL of digestion buffer (50 mM TEAB) were added onto the filter and were digested at 37 °C for 6 h. To elute peptides, three stepwise buffers were applied, with 100 μL of each with one more repeat, including 50 mM TEAB, 0.2% formic acid (FA), 50% acetonitrile, and 0.2% FA. The peptide solutions were pooled, lyophilized, and resuspended in 500 μL of 0.1 % FA.

Each sample was loaded onto individual Evotips for desalting and was then washed with 200 μL 0.1% FA, followed by the addition of 100 μL storage solvent (0.1% FA) to keep the Evotips wet until analysis. The Evosep One system (Evosep, Odense, Denmark) was used to separate peptides on a Pepsep column (150 μm inter diameter, 15 cm) packed with ReproSil C18 1.9 μm, 120A resin. The system was coupled to a timsTOF Pro mass spectrometer (Bruker Daltonics, Bremen, Germany) via a nano-electrospray ion source (Captive Spray, Bruker Daltonics). The mass spectrometer was operated in PASEF mode. The ramp time was set to 100 ms, and 10 PASEF MS/MS scans per topN acquisition cycle were acquired. MS and MS/MS spectra were recorded from *m*/*z* 100 to 1700. The ion mobility was scanned from 0.7 to 1.50 Vs/cm^2^. Precursors for data-dependent acquisition were isolated within ±1 Th and were fragmented with an ion mobility-dependent collision energy, which was linearly increased from 20 to 59 eV in positive mode. Low-abundance precursor ions with an intensity above a threshold of 500 counts but below a target value of 20,000 counts were repeatedly scheduled and otherwise dynamically excluded for 0.4 min.

### 2.6. Database Searching and Protein Identification

MS/MS spectra were extracted from raw data files and converted into .mgf files using MS Convert (ProteoWizard, version 3.0). Peptide spectral matching was performed with Mascot version 2.5 against the Uniprot human database. Mass tolerances were ±15 ppm for parent ions and ± 0.4 Da for fragment ions. Trypsin specificity was used, allowing for 1 missed cleavage. Met oxidation, Cys dioxidation, Cys thiol conversion to dehydroalanine, protein N-terminal acetylation, isopeptide bond formation with loss of ammonia (K), and peptide N-terminal pyroglutamic acid formation were set as variable modifications with Cys carbamidomethylation set as a fixed modification.

Scaffold (v 4.8, Proteome Software, Portland, OR, USA) was used to validate MS/MS-based peptide and protein identifications. Peptide identifications were accepted if they could be established at greater than 95.0% probability as specified by the Peptide Prophet algorithm. Protein identifications were accepted if they could be established at greater than 99.0% probability, and contained at least two identified unique peptides.

Oxidized cysteine content was determined by summing the spectral counts for peptides containing Cys sulfinic acid (Cys-SO2H, i.e., deoxidation) and Cys conversion to dehydroalanine, a stable end-product of oxidized cysteine intermediates. The irreversibly oxidized Cys content was normalized to total peptide spectral counts for the corresponding protein. Asparagine deamidation and glutamate/aspartate methylation were determined using database searches with these variable modifications. Results were normalized to total peptide spectral counts for the corresponding protein.

### 2.7. Statistics and Visualization

Statistical analysis was conducted using MetaboAnalyst v 5.0 with the following data normalizations: metabolomics data were sum-normalized, log transformed, and autoscaled; lipidomics data were sum-normalized and autoscaled; proteomics data were sum-normalized and autoscaled; Violin plots, bar graphs, and their corresponding statistics (one-way ANOVA with Kruskal–Wallis multiple comparisons test) were prepared in GraphPad Prism v 9.5.1. Network analysis and pathway analysis were performed in OmicsNet v 2.0, using as input the significantly altered metabolites and proteins (*p* < 0.05), the STRING database for protein–protein interactions, and the KEGG database for protein–metabolite interactions. The pathway analysis was performed in the Function Explorer node against the KEGG database.

## 3. Results

### 3.1. Clinical Presentation and Hematological Parameters

Two young Amish brothers presented at the Central Pennsylvania Clinic with severe disability from generalized dystonia with delays of gross motor milestones, poor head and trunk control, and uncontrolled movement of the extremities. The disability of the first boy was previously misdiagnosed as cerebral palsy attributed to an injury at birth. Genetic testing began when the second affected boy presented with similar symptoms. Recognized causes of Amish cerebral palsy were ruled-out by biochemical and targeted PCR-based mutation tests and a microarray-based genotype of parents for 1200-pathogenic variants in Lancaster County Amish population. This panel includes testing for many genetic causes of dystonia, spasticity, and ataxia, as well as by a 256-gene-panel of neurological disorders at InVitae, including glutaric aciduria type 1 (GA1), propionic acidemia, cobalamin-C and methylenetetrahydrofolate reductase (MTHFR) deficiencies, and congenital Parkinsonism. In retrospect, these screening panels did not include HPRT1. An HPRT1-variant of unknown significance was eventually found by whole-exome sequencing of the parents and two boys by two independent sequencing labs—Regeneron-Clinic for Special Children and NCGM by Neuberg—Central Pennsylvania Clinic. The affected male children inherited the single X-chromosome HPRT1 c.485 G>A;p.Ser162Asn variant, which their mother carries along with a normal allele.

After the HPRT1 variant was reported, serum uric acid levels were measured to help confirm the significance of the variant. A third affected infant in the family was diagnosed by targeted mutation testing of cord blood. Dystonia in the youngest child became apparent at 4–6 months of age; increased uric acid in the neonate was similar to that in the older siblings. At the time of initial work-up, none of the children displayed the self-injurious behavior characteristic of Lesch–Nyhan (LN). The middle boy began lip, tongue, and finger biting between 18 and 24 months of age. His older and younger siblings do not injure themselves.

The children’s mother is of Lancaster County Amish descent. She carries the X-linked variant in HPRT1 c.485 G>A;p.Ser162Asn, which appears to be a novel variant of unknown significance that structurally neighbors the binding site residues E134-T142 (Figure 1A,B) [2,18]. This variant is not seen in a database of more than 10,000 Amish whole exomes—a University of Maryland and Regeneron database—or in gnomAD (Broad Institute), which contains 250,000 exomes and whole genomes. No other cases of Lesch–Nyhan disease have been found in children of the mother’s sisters or her own mother’s generation. Inquiries by one of the authors (DHM) at Clinics for Amish Children with Genetic Disorders in Ohio, Kentucky, Indiana, Wisconsin, and Ontario, Canada did not uncover additional cases of the HPRT1 c.485 G>A;p.Ser162Asn mutation. We thus conclude that it is likely that this mutation is de novo in the boys’ mother. Whole-exome sequencing and dense microarray karyotype now allow the recognition of single family and single case de novo X-linked and autosomal-dominant mutations as well as rare autosomal recessive mutations and compound heterozygous cases that could not previously be found by homozygosity or genome-wide-SNP mapping.

This novel HPRT1-variant can now be classified as pathogenic and severe because of the neonatal onset of generalized dystonia and the emergence of self-injury in infancy. Increased uric acid is a biochemical feature of the disorder. Allopurinol is given to reduce the risk of uric acid kidney and bladder stones but the suppression of uric acid to normal levels does not improve the movement disorder or self-injury.

None of the three affected children has presented with overt anemia as judged by low hemoglobin or hematocrit (Table 1). One child has increased mean corpuscular volume (MCV) and red cell distribution width (RDW). The MCVs of the other two children are high normal but remain within the normal range. All three children had high serum uric acid concentrations at the time of diagnosis −5 to 6.5 mg/dL (normal level <2–4.5 mg/dL). Allopurinol suppressed serum uric acid to the normal range and reduced their risk of uric acid kidney stones; however, the normalization of serum uric acid did not improve their movement disorder.

### 3.2. Multiomics Signatures of LN in Patients with the HPRT1 c.485 G>A;p.Ser162Asn Mutation

In addition to the three affected children, their two parents voluntarily enrolled in this study: the father with normal HPRT and the mother who is heterozygous for HPRT deficiency (herein referred to as LN carrier). The family are members of the Pennsylvania Amish community, an underrepresented population in scientific research [19]. RBC samples were processed via mass spectrometry-based metabolomics, lipidomics, proteomics, and trace metal analysis (Figure 2A). Given the rarity of these specimens, two blood draws were obtained from each subject on the same day, then each sample was processed in technical triplicate (i.e., six measurements per subject). Comparing the LN RBCs (i.e., offspring) to the merged LN carrier and control (i.e., parents), we observed changes in particular to the metabolite and protein compartments (Figure 2B). Specifically, the two biomarkers of LN RBCs with the highest significance are the HPRT protein and its product GMP, both sharply decreased in LN.

High throughput metabolomics with a targeted approach to compounds in energy and redox metabolism yielded measurements of 192 metabolites (Appendix A). Principal component analysis (Figure 2C) suggests a differential metabolic phenotype of the LN RBCs in comparison to both the carrier and control RBCs, two groups whose profiles are more similar. Global changes in metabolite profiles are represented via heat map with hierarchical clustering of the top 40 metabolites by one-way ANOVA (Figure 2D). The metabolites affected in LN RBCs implicate numerous pathways related to nitrogen metabolism and illustrate the far-reaching effects of HPRT deficiency.

### 3.3. Dysregulation of Purine Metabolism Is a Hallmark of LN Patient RBCs, Only in Part Recapitulated in the Mutation-Carrying Mother’s RBCs

An overview of purine metabolism (Figure 2E) illustrates the effects of LN syndrome on RBCs and the degree to which allopurinol treatment normalizes purine levels in the two older siblings on the treatment. Expectedly, LN RBCs are deficient in HPRT at the protein level along with HPRT product GMP and upstream substrate guanine, though they have normal levels of IMP, as a result of the increased deamination of AMP in the three siblings but not in the mother carrying the mutation in heterozygosity without clinically relevant manifestations. In this view, it is interesting to note that LN carrier RBCs differ from control RBCs for several metabolites including HPRT product IMP (reduced in carrier) and substrate guanine (increased in carrier).

Other metabolic derangements include the elevation of hypoxanthine precursor and IMP dephosphorylation product—inosine. Importantly, allopurinol and its bioactive form oxopurinol are structural isomers of hypoxanthine and xanthine, respectively, and each pair of isobaric compounds coelutes due to the partners’ high degree of structural similarity (i.e., polarity). Though exact levels of hypoxanthine and xanthine cannot be discerned from the aforementioned isobaric drug and its catabolite, we observe that—in the two siblings on treatment—allopurinol normalizes RBC levels of urate in LN subjects. However, products downstream of urate, including 5-hydroxyisourate (5-OHisourate) and allantoate (non-enzymatic oxidation product in uricase-deficient human RBCs [20]), are less affected by the treatment with allopurinol.

### 3.4. Altered Glycolysis in LN RBCs Manifests with Dysfunctional Flux through Pyruvate Kinase

We next moved to an in-depth assessment of steady state levels of RBC energy metabolites. Though levels of glucose are not changed in LN RBCs, the depletion of hexose phosphate compounds and the accumulation of glycolytic intermediates downstream to glyceraldehyde 3-phosphate dehydrogenase (GAPDH) are observed in LN patients (Figure 3A). This signature becomes apparent beginning with bisphosphoglycerate (BPG, Figure 3A) through phosphoenolpyruvate (PEP). Allopurinol treatment appears to partially normalize the levels of these intermediates. There exists a sharp change in trend from PEP to pyruvate, whereby LN RBCs that possess abundant PEP also have significantly lower levels of pyruvate relative to the control and LN carrier RBCs. The conversion of PEP to pyruvate is catalyzed by the redox-sensitive enzyme [21] pyruvate kinase (PK), a rate-limiting and irreversible final step of glycolysis that catalyzes the second pay-off (ATP-generating) reaction. Though the PEP-to-pyruvate ratio is suggestive of a defect in PK expression or activity, changes in PK expression were not observed in the global proteomics analysis (Figure 3A). While deficiency in PK activity is not uncommon in the Amish Mennonite community [22], these patients do not carry any of the known PK mutations, which is suggestive of an oxidation-dependent inhibitory effect on this enzyme. However, post-translational modifications to active site cysteines that could explain a depressed PK catalytic activity—as is known in cancer and immune cells for various PK isoforms [21,23]—were not observed in this study. Glucose metabolism via the pentose phosphate pathway (PPP—Figure 3A), which is regulated by another enzyme encoded by a chromosome X-linked gene—glucose 6-phosphate dehydrogenase—was not broadly altered by HPRT status, though it is interesting to note that the RBCs of each of the three LN offspring had varying levels of the PPP end-product ribose phosphate. Interestingly, compared to the mother carrying the mutation, the LN siblings showed increases in the levels of 6-phosphogluconate, the entry step metabolite of the oxidative phase, NADPH-generating arm of the PPP—suggestive of moderate increases in the activation of antioxidant pathways in the LN group (Figure 3A).

### 3.5. RBCs from LN Patients Have Increased Free Fatty Acids and Acylcarnitines, a Hallmark of Altered Membrane Integrity and Deformability

Steady-state levels of many fatty acids are elevated in LN RBCs relative to carrier and control (Figure 3B, upper panel). In particular, the trend appears for fatty acyl chain lengths of C12 and longer, though C20:4 (i.e., arachidonic acid) is not changed and no significant increases in hydroxyeicosatetraenoic peroxidation products were observed in this group. The enrichment in fatty acids is more evident in the long chain poly- and highly unsaturated fatty acids (C18 and higher, 2 to 6 double bonds), and similar to glycolytic intermediates, only partially normalized by allopurinol. Acylcarnitines also trend toward elevation in the LN RBCs, though allopurinol more completely normalizes the levels of carnitines than RBCs, so few acylcarnitines have significant changes (Figure 3B, lower panel). Hydroxybutyrylcarnitine (C4-OH) had the greatest fold change difference for both LN and carrier vs. control though these values were also normalized with allopurinol treatment, suggestive of cross-talk between purine oxidation and membrane lipid remodeling.

### 3.6. LN Significantly Impacts Glutaminolysis, Conjugated Bile Acids and Tryptophan-Derived Inflammatory and Neurotransmitter Metabolites: Metabolic Signatures of Hepatic and Neural Dysfunction and Microbiome Dysbiosis

We next looked deeper into other metabolic pathways enriched in the top 40 heat map (Figure 2D). Figure 4A illustrates transamination substrates and products; we report in LN RBCs lower ratios of amide-containing amino acids glutamine and asparagine relative to their ester-containing counterparts glutamate and aspartate, respectively. In these subjects, there was not a measurable difference between LN carrier and control RBCs. Glutathione pools, including reduced (GSH) and oxidized (GSSG) forms, were elevated in RBCs from LN individuals (Appendix A). Urea cycle metabolites and polyamines were largely unchanged, except for metabolites downstream of arginine: citrulline and creatine are increased in the carrier and LN while guanidinoacetate and creatine are both decreased in the carrier and LN RBCs relative to control patient RBCs (Appendix A).

We observed low levels of methylation products, including guanidinoacetate, creatine, creatinine, choline, and glucosamine, as well as important precursors for purine nucleotides—asparagine, aspartate, glutamine, phosphate—and the depletion of the end products of purine synthesis—guanine, guanosine, GMP, GTP, and AMP. Substrates required to support purine de novo synthesis, including glycine and upstream metabolite serine, are not altered in LN RBCs though we note that samples from the LN individual not receiving allopurinol yielded six of the nine lowest glycine readings (Appendix A). Other requisite substrates, such as phosphoribosyl pyrophosphate (PRPP), were not detected.

Several bile acids (Figure 4B) are increased in LN RBCs, especially cholate- and taurine-conjugated bile acids, metabolites that are deconjugated by the gut microbiome and accumulate in the bloodstream as a result of inflammatory conditions triggering dysbiosis [24]. Given the neural manifestations of LN, it is interesting to note that RBCs harbor several of the enzymes that can synthesize neurotransmitters [25]—making the measurement of these compounds in blood cells a viable strategy to infer the dysregulation of such metabolic pathways in the central nervous system [10]. Here, we observed that adrenaline and dopamine are increased in LN RBCs—as opposed to results in murine models reporting decreases [26]—without changes to upstream essential amino acid tyrosine (Figure 4C). Similarly, while no changes were observed in the levels of the precursor tryptophan, serotonin levels were sharply decreased in both the carrier and LN RBCs, an observation that warrants additional investigation to the extent that dysfunctional serotonin synthesis and reuptake is etiologically linked to depression [27], and obsessive-compulsive disorders like facial grimacing are common manifestation of LN, including in the patients enrolled in this study. Of note, kynurenine levels were elevated in the sibling not on allopurinol, which is relevant in that this tryptophan metabolite is elevated in response to cGAS-STING-interferon activation secondary to infections (e.g., SARS-CoV-2). Kynurenine catabolism has been linked to adverse neurological manifestations in other populations with genetic anomalies, like subjects suffering from Trisomy 21 [28] or Aicardi–Goutieres syndrome [29].

Finally, to complete the small molecule analysis, we quantified levels of transition metals in RBCs from these subjects and report decreased levels of Fe, Zn, Se, and K in LN RBCs relative to both carrier and control (Figure 4D; see Appendix A for the full metallomics data set), which is directly relevant to erythropoiesis [4,30], redox biochemistry (including the synthesis of selenoproteins that counteract lipid peroxidation [31]), and neurological conditions [32].

### 3.7. The RBC Lipidome in LN Is Characterized by Elevation in Phosphatidylcholines and Depression in Sphingomyelins, Ceramides and Phosphatidylethanolamines

Untargeted lipidomics followed by database matching of MS^2^ data and identification yielded 257 named lipids, 103 of which were differentially abundant with *p* < 0.05 by one-way ANOVA. A PLS-DA of named lipids illustrates, similar to the metabolomics data, that the LN carrier RBCs have an intermediate lipid phenotype relative to the control and LN RBCs (Figure 5A). In fact, allopurinol treatment in two of the three LN subjects partially, yet incompletely, restores the lipid phenotype to levels compared to controls in the treated siblings relative to the youngest LN patient who is not receiving the drug. A heat map with hierarchical clustering of the top 50 lipids (by *p* value) reveals snapshot regions of the RBC lipidome that appear to be impacted most by HPRT heterozygosity and/or HPRT deficiency (Figure 5B). In particular, phosphatidylcholines (PCs) with highly unsaturated acyl chains are increased in LN RBCs (Figure 5C). Sphingomyelins, among others, are increased in the LN carrier but not in LN RBCs and thus warrant additional investigation to disentangle the effect of sex. These same SMs are lower in the two LN patients receiving allopurinol than in their untreated sibling. A view into the effects of allopurinol treatment on LN RBCs is presented via a volcano plot (Figure 5D) illustrating an allopurinol-associated increase in PCs and PEs and a corresponding decrease in SMs along with acylcarnitines (Figure 5D,E, here validated with unsupervised approaches on top of the targeted measurements described above in Figure 3B). Based on this analysis, we observed that the LN siblings presented an elevation in the RBC levels of phosphatidylcholines (PCs), accompanied by decreases in sphingomyelins, ceramides, and phosphatidylethanolamines (Figure 5E). 

### 3.8. The RBC Proteome in LN Is Characterized by Depletion of Structural Proteins, Elevation in the Levels of Acute Phase Response and Complement Proteins, and Elevated Protein Cysteine Oxidation

A global proteomics analysis of tryptic digests reveals 147 RBC proteins that were significantly altered, using one-way ANOVA (*p* < 0.05). RBCs from individuals with LN are characterized by a phenotypic shift evident on PLS-DA whereby RBCs from the LN carrier display an intermediate phenotype between the LN subjects and the control RBCs (Figure 6A,B). A hierarchical clustering analysis of the top 50 altered proteins (by *p* value) illustrates the protein expression changes that prominently characterize the LN RBC phenotype (Figure 6C). Specifically, HPRT peptides are undetectable in LN RBCs, in accordance with the complete lack of activity known in LN erythrocytes [33], and are higher in the carrier mother than in the control father (1.3-fold increase, *p* = 0.021), as anticipated for a chromosome X-encoded protein product. In addition, LN RBCs have decreased levels of hemoglobins, consistent with LN-associated anemia. It is crucial to note that RBCs in LN patients are typically macrocytic, thus hemoglobin copies per cell are increased; here, equal aliquots of RBCs reveal less hemoglobin in LN specimens compared to the carrier and control. In addition to hemoglobins, LN RBCs display decreased membrane and cytoskeletal proteins—dematin (DMTN), aquaporin 1 (AQP1), protein 4.1 (EPB41), Ras-related protein (RAB1B), and carbonic anhydrase 3 (CA3). We also observe a decrease in the LN RBCs of proteins involved in redox homeostasis, including 3-mercaptopyruvate sulfurtransferase (MPST)—an enzyme involved in the synthesis of cysteine—and H_2_S, aldehyde dehydrogenase (ALDH1A1)—which scavenges aldehydes produced by lipid peroxidation—and catalase (CAT), a key H_2_O_2_ detoxifying heme enzyme.

Conversely, a separate subset of the RBC proteome is enriched in LN subjects. Affected proteins include several related to heme and iron dynamics—uroporphyrinogen decarboxylase (UROD), ceruloplasmin (CP), hemopexin (HPX), and serotransferrin (TF)—and membrane structure and function—centractin (ACTR1A) and fibronectin (FN1). Additionally, purine salvage enzyme adenine phosphoribosyltransferase (APRT), which produces AMP, is elevated in LN subjects perhaps as a compensation for dysregulation induced by the absence of functional HPRT. Finally, we observed elevated levels of complement system proteins (C4BPA, CFH, CP, CFB) along with increases in proteins involved in coagulation and hemostasis, such as the serpins, fibrinogens, kininogen 1 (KNG1), and prothrombin (F2). These acute phase response proteins have been previously associated with the results of inflammatory events that promote deposition on the RBC membrane and prime the RBC for increased susceptibility to intra- and extra-vascular hemolysis [34], especially in response to interferon-triggering infections like SARS-CoV-2 [17].

As anticipated above for PK, RBC metabolism is modulated in part by disruptions to redox homeostasis. A particular challenge in RBCs is the inability to synthesize proteins de novo (lack of organelles), thus RBCs are uniquely susceptible to oxidative damage and reliant on repair/degradation pathways. We assessed levels of modified proteins by quantifying (a) the oxidation of protein cysteines to form sulfinic acid (P-SO_2_H) and the elimination of sulfur to form protein dehydroalanine (DHA); (b) carboxylate side chain methylation (Glu, Asp) along with asparagine deamidation followed by methylation of the L-isoaspartyl groups resulting from this reaction (Figure 6D).

With respect to oxidized cysteine, we identified 16 proteins in the global proteomics analysis with a measurable formation of these oxidized cysteine products (Figure 6E). Spectral counts of peptide sulfinic acids and peptide dehydroalanine were summed then subsequently normalized to total peptide levels (i.e., spectral counts) for the corresponding protein. As displayed in Figure 6E, the majority of these proteins are unexpectedly less oxidized (at cysteine) in the LN RBCs than in carrier or control (i.e., levels < 1). The few notable exceptions are catalase (CAT), the key heme-containing H_2_O_2_ detoxifying enzyme, methanethiol oxidase (SELENBP1), glycolytic enzyme and known ROS target glyceraldehyde 3-phosphate dehydrogenase (GAPDH) [35], and spectrin alpha (SPTA1). Of these, there is more oxidized GAPDH in LN RBCs relative to carrier and control; for the remainder, LN RBCs have lower levels of oxidized cysteine-containing peptides.

Oxidant stress in RBCs leads to damage of asparagine and glutamine residues by deamidation followed by methylation of the resulting isoaspartate and isoglutamate residues [36]. The methylation of (iso)aspartate and glutamate side chains was observed in hemoglobin gamma (HBG1), complement C4-B (C4B), albumin (ALB), and alpha-2-macroglobulin (A2M) at the highest levels in LN RBCs versus carrier and control (Figure 6F). In contrast, only catalase (CAT) has the highest methylated Asp and Glu content in the carrier. Here, we report elevated N-deamidation/methylation occurring in LN RBCs on β-actin (ACTB), purine nucleoside phosphorylase (PNP), SPTA1, adenosine 5′-monophosphoramidase (HINT1), and protein 4.1 (EPB41)—a marker of RBC aging in vivo—relative to both carrier and control (Figure 6G). Levels of this modification in ACTB, PNP, and EPB41 were lowest in the control. These PTM observations warrant additional experimentation, particularly if the RBC lifespan is different in LN subjects versus the general population. A shortened RBC lifespan would be expected to result in a greater proportion of young RBCs in circulation, which would possess fewer age-associated oxidation events.

Finally, metabolomics and proteomics data sets were integrated into an interaction network (Figure 6H), revealing links among: (a) nucleotides and HPRT (green dots); (b) redox metabolism (red dots); and (c) energy metabolism (blue and purple dots). A pathway analysis against the KEGG database supports the alterations of these pathways in LN RBCs (Figure 6I).

## 4. Discussion

Herein, we report the first integrated multi-omics investigation of how Lesch–Nyhan disease caused by an HPRT1 c.485 G>A;p.Ser162Asn variant impacts the molecular composition of RBCs. Without means to synthesize ATP de novo, RBCs are reliant on the purine salvage pathway to restore ATP pools. Though hypoxanthine is not a direct substrate for purine salvage production of ATP, enzymopathies in the pathway disrupt metabolic homeostasis; in the case of Lesch–Nyhan disease, HPRT deficiency results in the underutilization of hypoxanthine and xanthine, leading to the accumulation of urate and purine monophosphates. Disruptions to the purine salvage pathway in RBCs have been noted in the context of iatrogenic or environmental stimuli that trigger oxidant stress to the erythrocyte, such as blood storage [11], irradiation, exercise [37,38], cryostimulation [39], propionic acidemia, exposure to hypoxia ex vivo and in vivo, and Down syndrome (i.e., trisomy 21) [40].

Here, we leverage a unique and rare sample set voluntarily donated by a family carrying a novel S162N HPRT mutation to perform a comprehensive multiomics analysis. Global metabolomics profiles suggest that the HPRT deficiency drives the RBC phenotypic changes; far fewer differences are observed when comparing the control (HPRT normal) to the carrier RBCs, which is consistent with the lack of clinical manifestations in the mother carrying the mutation. The proteomics characterization also abides by this theme in that control and carrier are similar to each other and LN RBCs offer a shifted phenotype. In contrast, global lipidomic profiles reveal marked differences between the control (father) and carrier (mother) RBCs (Figure 4B, bottom half). With low n, it is not possible to disentangle the effect of sex alone on the observed changes, though it is important to note that the two subjects are similar in age.

Though with few subjects, given the rarity of this mutation, the treatment of two of the three LN individuals with allopurinol allows for the preliminary insight into how such treatment impacts RBCs and where dysregulated metabolic phenotypes are restored, even if partially. For example, RBC free fatty acids and acylcarnitine levels are partially normalized in LN patients receiving allopurinol, confirming prior reports about a crosstalk between purine oxidation and activation of the Lands cycle [41,42,43]. The proteomics profiles of RBCs from allopurinol-treated subjects suggest a decrease in disease severity, though the effect is relatively modest (Figure 5C). Lipidomics profiles are unique in that LN RBCs display significant differences that appear to be imparted by allopurinol treatment, in particular among the phosphatidylcholines (PCs) and sphingomyelins. Interestingly, PC-derived choline could contribute methyl groups to fuel protein isoaspartyl methylation upon deamidation of asparagine residues, a protein damage repair pathway that is regulated by PIMT in mature RBCs [36]. Interestingly, genetic ablation of PIMT is lethal in mice and—like LN—manifests with neurological sequelae (seizures at 6–8 weeks of age when in homozygosity [44], accelerated aging of the brain when in heterozygosity [45]) as well as an RBC oxidant stress phenotype [36]. In this view, it is interesting to speculate that deamidation of N162 in the novel mutant HPRT investigated in this study would restore the negative charge of the serine residue in the dominant canonical HPRT, provided that isoaspartyl formation does not rearrange the HPRT protein backbone and further compromise its enzymatic activity. This observation is relevant in that it suggests that the supplementation of methyl group donors (e.g., choline, betaine, vitamin B5, *S*-adenosylmethionine) may partially restore the phenotype observed in this patient population as a co-adjuvant to allopurinol treatment.

Though only observed in one of the three children from the small patient cohort in this study, the macrocytic phenotype common in individuals with LN results from dysfunctional one-carbon metabolism—critical to hematopoiesis [40], a common trait between LN and another genetic condition accompanied by macrocytic anemia—Down syndrome [46]. Interestingly, elevated kynurenine in Trisomy 21 is associated with the accumulation of its neurotoxicant catabolites [28], another phenotypic overlap between the LN patients investigated here and Down syndrome. In this view, it is worth noting that the activation of the interferon-indole 2,3-dioxygenase (IDO)-kynurenine axis is common among LN, Down syndrome, and Aicardi–Goutieres syndrome, also associated with clinical neurological manifestations [29,47]. Similarly, the elevated synthesis of neurotoxic kynurenine metabolites is a feature of glutaric aciduria type 1 (GA1). Clinical parallels exist between Lesch–Nyhan and glutaric aciduria 1 (GA1), a pathology caused by the incomplete catabolism of lysine and tryptophan. For example, in both disease states infants appear normal at birth and the movement disorder emerges over the first 1–2 years of life. Over the 35-year period of 1988–2023, GA1 research provided insights that have led to increasingly effective, preventative treatments. The outcomes of affected infants have improved from 95% disabled to 95% with good neurological outcomes [48]. Metabolic intervention in GA1 consists of a low lysine diet with supplementation of a lysine-free, tryptophan-reduced, amino acid mixture, and oral supplementation of L-carnitine. Similar metabolic interventions could be tested to mitigate the neurological manifestations of LN.

Since the kynurenine pathway is downstream of the cGAS-STING viral/pathogen sensing pathway, this observation could contribute to explaining why disease severity worsens in LN patients upon infection [49,50,51,52]. Finally, it is interesting to speculate that increased reticulocytosis or mitochondrial-containing mature erythrocytes—as it is observed for example in sickle cell patients [53] or lupus erythematosus [54]—results in an increased basal activation of humoral immune responses downstream of interferon signaling . Whether this phenomenon occurs in LN is not clear, though our data clearly point at the elevation of carboxylic acids and mitochondrial proteins in circulating RBCs.

Severe depletion of serotonin in both LN patients and the mutation-carrying mother is consistent with elevated tryptophan catabolism towards the kynurenine pathway at the expense of the synthesis of this important metabolite, an additional factor that contributes to explaining the neurological disorder in the LN population.

We are not aware of other studies that have investigated the impact of allopurinol on the RBC lipidome, which could be translationally relevant for patients with gout; such characterizations are warranted, as dynamic changes to the lipidome could alter RBC membrane structure and therefore cellular function. Despite benefits of purine catabolism, energy metabolism—especially the payoff steps of glycolysis downstream to pyruvate kinase—appeared depressed in the LN patients, suggestive of a potential benefit to the use of PK activators (e.g., mitapivat [55]) as a co-adjuvant in the treatment of LN.

## 5. Conclusions

In conclusion, this multiomics investigation provides a wealth of heretofore unknown molecular level information about how HPRT deficiency impacts the RBC and allows for the contextualization of the rare Lesch–Nyhan disease alongside other diseases and physiological changes (e.g., exercise) that impact purine metabolism. We recognize as a limitation of this study the limited size of the patient cohort. Further, we acknowledge that, though various HPRT mutations are known to result in LN disease, the molecular profiles described here are derived from RBCs specifically carrying the HPRT1 c.485 G>A;p.Ser162Asn mutation. The data herein could inform future complementary treatment to present regimens like allopurinol, for example, via supplementation of substrates to fuel compensatory regulatory pathways, here observed to be active in LN patients and the healthy carrier of this mutation, such as protein isoaspartyl methylation.

## Figures and Tables

**Figure 1 antioxidants-12-01699-f001:**
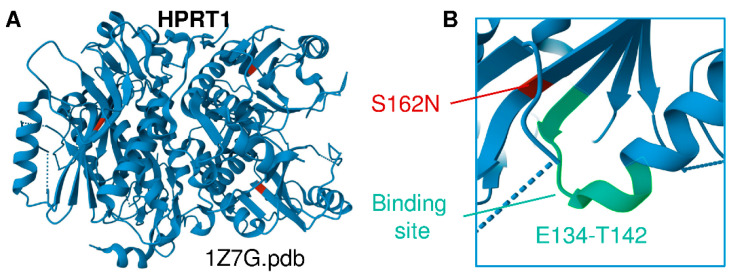
(**A**) Structural representation of HPRT1 (PDB ID 1Z7G). (**B**) Location of the Ser162Asn mutation identified in this cohort, a previously unreported HPRT1 mutation that is adjacent to the substrate binding site (E134-T142 region).

**Figure 2 antioxidants-12-01699-f002:**
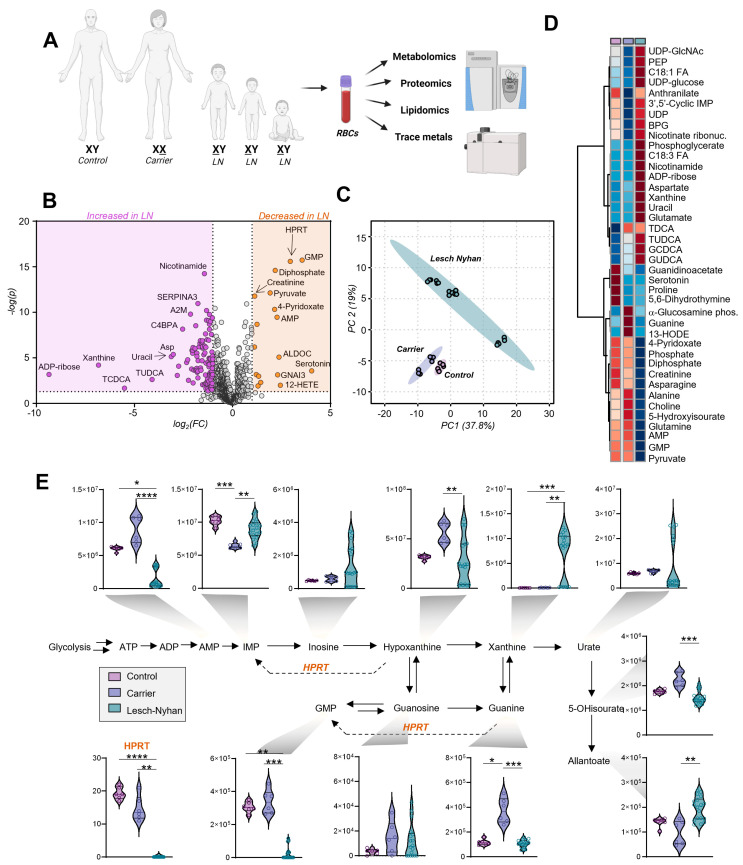
Multiomics analysis of red blood cells (RBCs) from 3 individuals with Lesch–Nyhan disease and their parents—a heterozygous carrier mother and clinically healthy control father. (**A**) Omics study design; (**B**) Volcano plot of all omics results, LN vs. nonLN. Orange indicates molecules significantly elevated in LN relative to carrier and control, purple indicates significantly decreased molecules in LN. (**C**) Principal component analysis (PCA) of metabolomics results; (**D**) Hierarchical clustering analysis of the top 40 metabolites by ANOVA. Group means are shown. (**E**) Purine metabolism and salvage. Data are peak areas (au), statistics are one-way ANOVA with Kruskal–Wallis multiple comparisons test. * *p* < 0.05, ** *p* < 0.01, *** *p* < 0.001, **** *p* < 0.0001.

**Figure 3 antioxidants-12-01699-f003:**
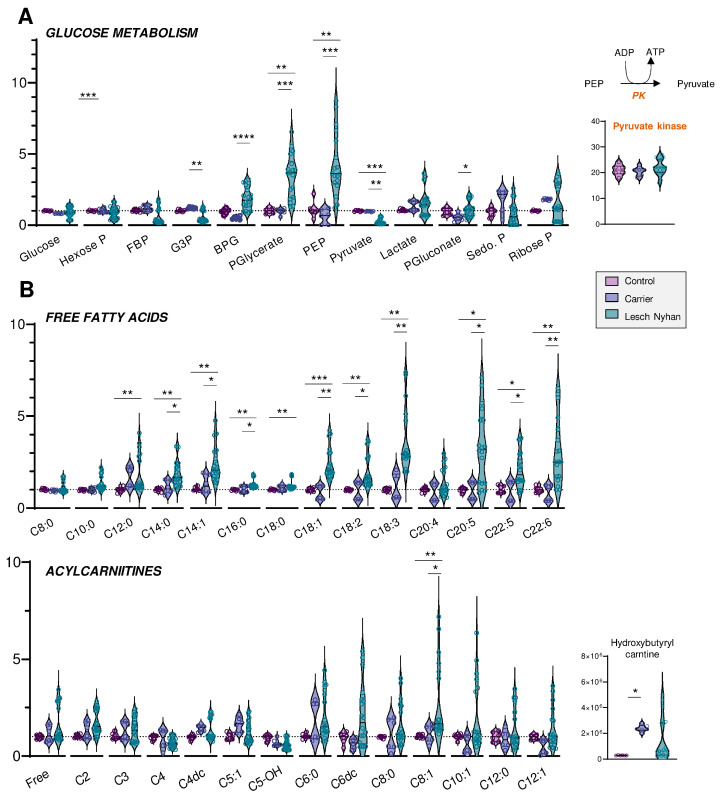
(**A**) Overview of glycolysis and the pentose phosphate pathway. (**B**) Levels of free fatty acids (**top**) and acylcarnitines (**bottom**) by increasing acyl chain length. Data are presented normalized to control samples. Statistics are one-way ANOVA with Kruskal–Wallis multiple comparisons test. * *p* < 0.05, ** *p* < 0.01, *** *p* < 0.001, **** *p* < 0.0001.

**Figure 4 antioxidants-12-01699-f004:**
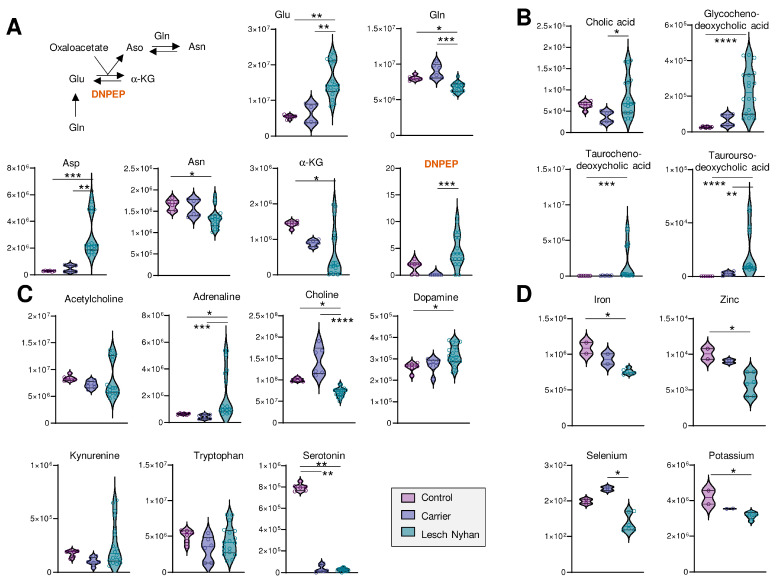
(**A**) Overview of transamination reactions reveals decreased amido-containing amino acids Gln and Asn in LN RBCs and increased carboxylate amino acids Glu and Asp. (**B**) RBC bile acids are elevated in LN subjects’ RBCs. (**C**) Levels of neurometabolites. (**A**–**C**) Data are peak areas (au). (**D**) Trace metal analysis (ppb). Statistics are one-way ANOVA with Kruskal–Wallis multiple comparisons test. * *p* < 0.05, ** *p* < 0.01, *** *p* < 0.001, **** *p* < 0.0001.

**Figure 5 antioxidants-12-01699-f005:**
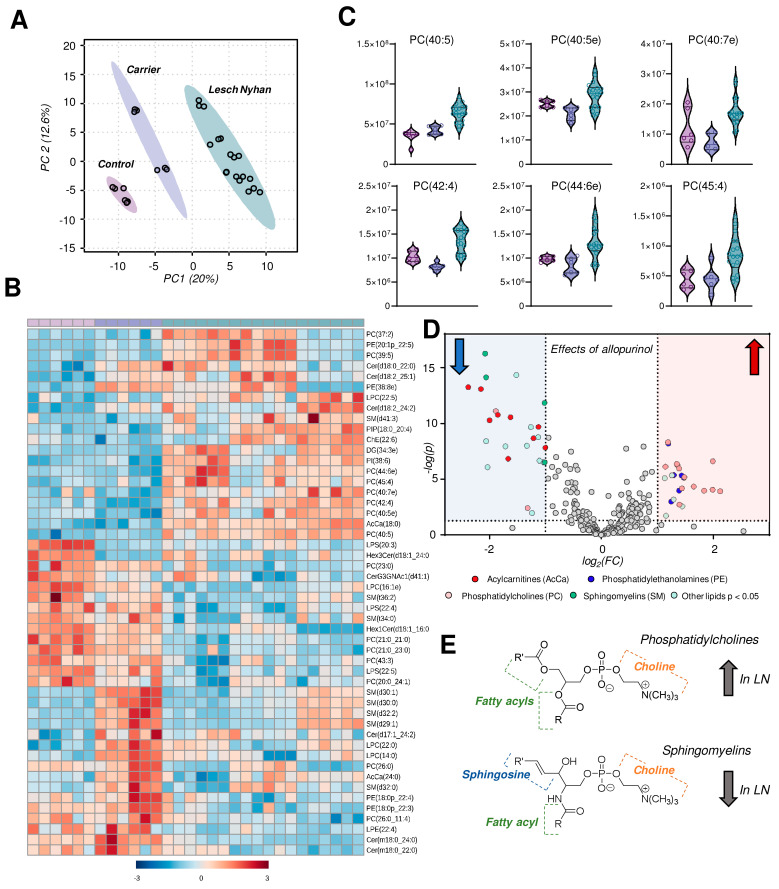
Lipidomics profiles of RBC samples from Lesch–Nyhan subjects versus carrier and control. (**A**) Partial least squares-discriminant analysis (PLS-DA) reveals a clustering of samples along component 1 (x-axis) illustrating that LN carrier RBCs possess an intermediate phenotype between LN and control RBCs. (**B**) Heat map with hierarchical clustering of the top 50 metabolites by one-way ANOVA. (**C**) Violin plots of highly unsaturated phosphatidylcholines. (**D**) Volcano plot of the effects of allopurinol treatment (2 LN subjects) versus untreated (1 LN subject) on the RBC lipidome. The red shaded region indicates lipids that are elevated in allopurinol-exposed RBCs; the blue shaded region indicates lipids that are decreased in allopurinol-exposed RBCs. (**E**) General structures of phosphatidylcholines (**top**) and sphingomyelins (**bottom**).

**Figure 6 antioxidants-12-01699-f006:**
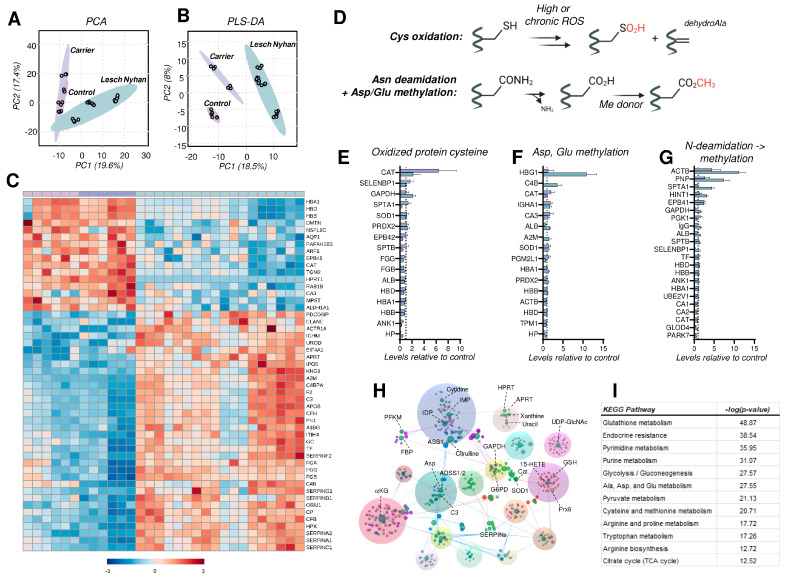
Proteomic profiling of RBCs from Lesch–Nyhan subjects. (**A**) Unsupervised principal component analysis demonstrates overlapping phenotypes in carrier and control RBCs. (**B**) Partial least squares-discriminant analysis illustrates the clustering of biological groups and the intermediate phenotype of the carrier RBCs. (**C**) Global proteomics results as visualized via heat map with hierarchical clustering of the top 50 proteins by *p*-value (one-way ANOVA). (**D**) Schematic of post-translational modifications (PTMs) investigated as readouts for proteome oxidation, including cysteine oxidation, asparagine deamidiation, and methylation of glutamate and aspartate. (**E**) Irreversibly oxidized protein cysteine (sulfinic acid and dehydroalanine) content as a function of total protein levels. Levels are presented relative to control RBCs (dashed line at x = 1). (**F**) Methylation of protein glutamate and aspartate as a function of total protein levels. Levels are presented relative to control RBCs (dashed line at x = 1). (**G**) Asparagine deamidation with subsequent aspartate methylation catalyzed by PIMT1 and other methylating enzymes. Levels are presented relative to control RBCs (dashed line at x = 1). (**H**) Network analysis of protein-protein and protein-metabolite interactions. Labeled note color denotes metabolic pathway: green—nucleotides, blue—amino acids, purple—energy metabolism, red—redox metabolism. (**I**) Pathway analysis of integrated proteomics and metabolomics data. Abbreviations found in Appendix A.

**Table 1 antioxidants-12-01699-t001:** Complete blood count (CBC) parameters for the three enrolled patients with c.485 G>A;p.Ser162Asn HPRT mutation. Abbreviations: WBC = white blood cell × 10^9^/L, RBC = red blood cell in 10^6^/mL, Hgb = hemoglobin beta in g/dL, Hct = hematocrit in %, MCV = mean corpuscular volume in fL, MCH = mean corpuscular hemoglobin in pg, MCHC = mean corpuscular hemoglobin concentration in g/dL, platelet count × 10^9^/L, RDW = red cell distribution width in fL.

	**Patient 1**		**Patient 2**			**Patient 3**	
**Age at draw (yrs)**	*4.25*	*5.00*	*2.75*	*3.25*	*3.67*	*0.75*	*1.50*

**WBC count**	7.3	5.9	7.8	8.6	6.0	6.2	5.6
**RBC count**	4.03	4.05	4.23	4.18	3.89	4.58	4.29
**Hgb**	12.5	12.2	12.4	12.7	12.9	12.3	12.5
**Hct**	37.6	38.6	37.6	40.0	39.3	37.1	39.1
**MCV**	93.3	95.3	88.9	95.7	101.0	81	91.1
**MCH**	31.0	30.1	29.3	30.4	33.2	26.9	29.1
**MCHC**	33.2	31.6	33.0	31.8	32.8	33.2	32.0
**Platelet count**	308	337	300	298	318	310	310
**RDW**	14.4	15.9	15.2	14.3	14.5	15.4	17.8

Data in red and blue indicate values that are above or below normal range, respectively.

## Data Availability

Data generated in this study are available in Appendix A and/or from the authors upon reasonable request.

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
