# Peer review of "Red Blood Cells from Individuals with Lesch–Nyhan Syndrome: Multi-Omics Insights into a Novel S162N Mutation Causing Hypoxanthine-Guanine Phosphoribosyltransferase Deficiency"

_antioxidants, 2023, doi:10.3390/antiox12091699_

Round 1

Reviewer 1 Report

This paper by Reisz et al describes the metabolic abnormalities of red blood cells from a family with Lesch-Nyhan syndrome due to a novel loss of function mutation in the gene encoding the hypoxanthine-guanine phosphoribosyl transferase gene (HPRT1). The new HPRT1 mutation linked with Lesch-Nyhan syndrome is the most novel and interesting information provided by the paper. It is strongly recommended to change the title to highlight this novelty.

Based on the location close to the substrate-binding site of the protein, the authors suggest that the mutation is loss of function. However, the paper does not include any functional data (metabolism of cell lines genetically engineered to carry the mutation, etc etc) that are usually provided to formally support the casual link between a new mutation and a disease. This weakness is partially overcome by the multi-omics analyses of the red blood cells from the blood of the patients. This analysis clearly indicates that the red blood cells from the kindreds have a HPRT1 deficiency signature, overcoming in an inventive way the lack of formal experiments described above.

The experiments are well designed and the results are strong. The insufficient power of the comparison between the diseased and healthy controls is appropriately discussed.

Problems are found in the interpretation and discussion of the data. Instead of discussing their data in terms of a novel mutation, the authors decide to use them to explain the macrocytic anemia associated with the Lesch-Nyhan syndrome. This discussion is weak because macrocytic anemia usually reflects accelerated maturation at the erythroid progenitor cell level and not the effects of hypoxia at the red blood cell level. Without knowing what happens at the progenitor cell levels, the link between the data and the red blood cell phenotype of the patients is unclear. It is strongly recommended to increase the focus of the discussion of a novel mutation and to decrease that on macrocytic anemia.

Specific comments

The blood counts of the individuals included in the study should be disclosed to prove that the kindreds have macrocytic anemia.

Line 80: “The three brothers were from an Amish Family”. This information is irrelevant for the interpretation of the data and it may be perceived as a breach of confidentiality. It should be removed.

Author Response

Reviewer 1: This paper by Reisz et al describes the metabolic abnormalities of red blood cells from a family with Lesch-Nyhan syndrome due to a novel loss of function mutation in the gene encoding the hypoxanthine-guanine phosphoribosyl transferase gene (HPRT1). The new HPRT1 mutation linked with Lesch-Nyhan syndrome is the most novel and interesting information provided by the paper. It is strongly recommended to change the title to highlight this novelty.

Based on the location close to the substrate-binding site of the protein, the authors suggest that the mutation is loss of function. However, the paper does not include any functional data (metabolism of cell lines genetically engineered to carry the mutation, etc etc) that are usually provided to formally support the casual link between a new mutation and a disease. This weakness is partially overcome by the multi-omics analyses of the red blood cells from the blood of the patients. This analysis clearly indicates that the red blood cells from the kindreds have a HPRT1 deficiency signature, overcoming in an inventive way the lack of formal experiments described above. The experiments are well designed and the results are strong. The insufficient power of the comparison between the diseased and healthy controls is appropriately discussed.

Problems are found in the interpretation and discussion of the data. Instead of discussing their data in terms of a novel mutation, the authors decide to use them to explain the macrocytic anemia associated with the Lesch-Nyhan syndrome. This discussion is weak because macrocytic anemia usually reflects accelerated maturation at the erythroid progenitor cell level and not the effects of hypoxia at the red blood cell level. Without knowing what happens at the progenitor cell levels, the link between the data and the red blood cell phenotype of the patients is unclear. It is strongly recommended to increase the focus of the discussion of a novel mutation and to decrease that on macrocytic anemia.

Authors’ reply: Thank you for the kind and constructive summary of the study. Following this reviewer’s suggestion, we revised the title to highlight the novel mutation. We also revised the discussion section to limit the focus on the hematological complications and rather provide a more detailed description of the clinical manifestation of LN in the children who were part of this study.

Specific comments:

The blood counts of the individuals included in the study should be disclosed to prove that the kindreds have macrocytic anemia.

Authors’ reply: We have added a table (Table 1) of blood count information using CBC results (2-3 per patient) available to us along with accompanying text in the Results section.

Line 80: “The three brothers were from an Amish Family”. This information is irrelevant for the interpretation of the data and it may be perceived as a breach of confidentiality. It should be removed.

Authors’ reply: Thank you for this suggestion; this particular instance has been removed. Other mentions of the broader Amish (Plain people) community such as in the methods section and at the beginning of the Results are relevant to the occurrences of rare genetic mutations and altered health risks arising in genetically isolated communities.

Reviewer 2 Report

This is a very well written report of careful analysis of multi-omic investigation of RBCs in Lesch-Nyhan syndrome (LNS). So far as I can tell, RBCs in LNS have only received limited investigation, and that was in the 1970s and only two or three papers then.

The high quality of the figures is exemplary.

The brief comments on the potential contribution of abnormalities in one-carbon metabolism to macrocytosis are thought-provoking.

1. If the information is available to the authors, it would be helpful to have a brief characterization of the clinical phenotype of the LNS patients in the initial Materials & Methods section. It would be particularly significant to include the degree of macrocytosis and of anemia if present, and if possible the red blood cell indices particularly the mean corpuscular hemoglobin (MCH).

2. In the proteomics discussion, the authors note that the amount of RBCs hemoglobin is decreased in the patients. That would be expected since presumably a similar quantity of blood was obtained from patients, carrier, and control, and the patients would presumably be anemic (less hemoglobin per unit volume of blood). Since LNS patients are typically macrocytic, the amount of hemoglobin per individual RBC is typically elevated – there are just fewer RBCs. The authors should indicate that this decreased hemoglobin on their analysis reflects hemoglobin in the total sample and not (or at least not necessarily) the amount of hemoglobin per individual RBC.

Author Response

Reviewer 2: This is a very well written report of careful analysis of multi-omic investigation of RBCs in Lesch-Nyhan syndrome (LNS). So far as I can tell, RBCs in LNS have only received limited investigation, and that was in the 1970s and only two or three papers then. The high quality of the figures is exemplary. The brief comments on the potential contribution of abnormalities in one-carbon metabolism to macrocytosis are thought-provoking.

  1. If the information is available to the authors, it would be helpful to have a brief characterization of the clinical phenotype of the LNS patients in the initial Materials & Methods section. It would be particularly significant to include the degree of macrocytosis and of anemia if present, and if possible the red blood cell indices particularly the mean corpuscular hemoglobin (MCH).

Authors’ reply: We have now included a) additional brief description in the methods section re clinical presentation; b) more thorough narrative of clinical encounters and timeline at the beginning of the Results section; c) a new table (Table 1) of CBC information.

  1. In the proteomics discussion, the authors note that the amount of RBCs hemoglobin is decreased in the patients. That would be expected since presumably a similar quantity of blood was obtained from patients, carrier, and control, and the patients would presumably be anemic (less hemoglobin per unit volume of blood). Since LNS patients are typically macrocytic, the amount of hemoglobin per individual RBC is typically elevated – there are just fewer RBCs. The authors should indicate that this decreased hemoglobin on their analysis reflects hemoglobin in the total sample and not (or at least not necessarily) the amount of hemoglobin per individual RBC.

Authors’ reply: We appreciate the insight and have added content in the proteomics results section to reflect this consideration.